# Existential Suffering in Palliative Care: An Existential Positive Psychology Perspective

**DOI:** 10.3390/medicina57090924

**Published:** 2021-09-01

**Authors:** Paul T. P. Wong, Timothy T. F. Yu

**Affiliations:** 1Department of Psychology, Trent University, Peterborough, ON K9L 0G2, Canada; 2Department of Psychology, University of Toronto, Toronto, ON M1C 1A4, Canada; timothy.tfm.yu@gmail.com

**Keywords:** palliative care, meaning therapy, CALM therapy, COVID-19, existential positive psychology, good death, wellbeing, mature happiness, flourishing

## Abstract

The COVID-19 pandemic has exposed the inadequacies of the current healthcare system and needs a paradigm change to one that is holistic and community based, illustrated by the healing wheel. The present paper proposes that existential positive psychology (PP 2.0) represents a promising approach to meet the rising needs in palliative care. This framework has a twofold emphasis on (a) how to transcend and transform suffering as the foundation for wellbeing and (b) how to cultivate our spiritual and existential capabilities to achieve personal growth and flourishing. We propose that these objectives can be achieved simultaneously through dialectical palliative counselling, as illustrated by Wong’s integrative meaning therapy and the Conceptual Model of CALM Therapy in palliative care. We then outline the treatment objectives and the intervention strategies of IMT in providing palliative counselling for palliative care and hospice patients. Based on our review of recent literature, as well as our own research and practice, we discover that existential suffering in general and at the last stage of life in particular is indeed the foundation for healing and wellbeing as hypothesized by PP 2.0. We can also conclude that best palliative care is holistic—in addition to cultivating the inner spiritual resources of patients, it needs to be supported by the family, staff, and community, as symbolized by the healing wheel.

## 1. Introduction

The COVID-19 pandemic has exposed the inadequacies of palliative care services with over 4 million deaths and 100 million confirmed cases. All healthcare workers, including palliative care workers, have faced severe challenges such as shortage of beds and staff, long working hours, and a lack of personal protective equipment [1,2,3]. The Toronto palliative care situation illustrates the need to integrate palliative care into COVID-19 management, and to optimize it for the pandemic [4].

Demographic trends also demand strategic thinking and planning in order to meet the challenge of increased demands for palliative care because of increased longevity accompanied by an increase in psychological needs, such as meaning for living, will to live, and death acceptance [5,6,7]. The existential crisis is especially severe for those who are approaching the end of life with cumulative losses in all areas of life.

For end-of-life care, the best medicine is compassion: “The word compassion means ‘to suffer with.’ Compassionate care calls physicians to walk with people in the midst of their pain, to be partners with patients rather than experts dictating information to them.” [8]. “We are at our best, when we serve each other”, wrote Byock [9], one of the foremost palliative care physicians in the US. He argued that the healthcare system should not be dominated by high-tech procedures and a philosophy to “fight disease and illness at all costs.” To ensure the best possible elder care, we must not only remake our healthcare system, but also move beyond our cultural aversion to thinking about death.

### 1.1. A Holistic Model of Compassionate Care

The current pandemic reminds me (the senior author) of the SARS crisis, which also exposed the inadequacy of a healthcare system based on the well-entrenched bio-medical model. The courage and personal sacrifice of frontline healthcare workers and volunteers in the face of death demonstrated the crucial role of the spiritual dimension, such as compassion, self-transcendence, and existential courage, which was my focus in my keynote on compassion, funded by the medical authority in Hong Kong [10].

By tapping into spiritual resources, my holistic model (see Figure 1) aimed to improve healthcare services without a corresponding increase in cost. The setting for my speech in Hong Kong—the historical Alice Ho Miu Ling Nethersole Hospital—served as a symbol of a spiritually oriented holistic healthcare. For more than 100 years, Nethersole Hospital earned a reputation of consistently providing compassionate quality care for the residents of Hong Kong; their mission statement was: “To bring Life to Mankind in its fullness through enhancement of Wellness of the Total Person and Compassionate Care of the Sick”.

The Healing Wheel is still relevant today. First of all, to be one’s best, the healer (the healthcare provider) needs to be spiritually connected with God (or a higher power) and becomes transformed through a set of religious beliefs or rituals. As a spiritually transformed person, they must be a conduit for spiritual blessings. Their love and faith will have an impact on their patients and the healing community. When they work with the patients, they are praying for wisdom, strength, and healing.

Compassionate human encounters in a supportive and caring environment can be a powerful source of healing. For the healers, they find meaning and purpose in serving their God/gods and serving others. For the patients, they rediscover the meaning of hope and love through the compassionate care they have received. For the community, their voluntary acts of altruism and compassion bring blessings not only to the patients but also to themselves [11,12].

### 1.2. Why Is Existential Positive Psychology (PP 2.0) Important for Palliative Care

The above holistic healthcare model can be best understood and practiced from the existential positive psychology (EPP) perspective, also known as the second wave positive psychology (PP 2.0) [13], as distinguished from the positive psychology launched by Seligman [14]. The obvious meaning of EPP is that it integrates the ultimate concerns of existentialism with the universal desires for happiness and wellbeing emphasized by positive psychology. Thus, by definition, EPP integrates the positive potential and the dark side of human existence, resulting in resilience and mature happiness. The main thrust of EPP is the positive transformation of life as a whole to complete the circle of wellbeing [15,16].

The theoretical and practical implications of EPP are numerous [15,17,18]. In fact, according to Wong’s latest thinking [19], most of human suffering, from developmental crisis to inner struggles, belong to the category of existential suffering. Therefore, suffering is considered the necessary foundation for wellbeing, and wellbeing can be achieved only through the dialectical process of the dual-system model [20] or Yin–Yang interactions, as illustrated in Figure 2. 

This above model explains how the new science of resilience and flourishing through transforming suffering works. Firstly, life is seldom smooth sailing due to our inherent limitations and foibles, the inevitable suffering at every stage of human development, and external disruptive events, such as the pandemic, wars, and natural disasters. Often, suffering is an inescapable aspect of life. For instance, Salman Akhtar [21] considers fear, greed, guilt, deception, betrayal, and revenge as the six major sources of suffering, which cannot be eliminated by the medical or cognitive models. Nor can they be eliminated by seeking happiness, because the pursuit itself may be a source of suffering from greed and disappointment. That is why the positive psychology of focusing on the pursuit of happiness does not work, especially during the pandemic. Secondly, this new paradigm proposes that wellbeing depends on the approach and avoidance systems working together, any setback during the pursuit of happiness will trigger the aversive system, and in coping with inevitable suffering, we are capable of transforming it into strength and joy through personal growth. Thus, by embracing suffering, the approach system has a better chance of success, and the avoidance system has a better chance of transforming suffering into wellbeing.

From the perspective of existential positive psychology [13,14], all emotions, including painful ones, have adaptive value because they help enhance our resilience, meaning, and flourishing. Paradoxically, death holds the key to living a vital, authentic, and meaningful life [22]. According to Yalom [23], although the physicality of death may destroy us, the idea of death has saved many lives. The challenge for PP 2.0 is how to transform death anxiety into death acceptance, a life of significance, and mature happiness.

The positive psychology of death anxiety can be best understood in terms of the dual-system model [20]. According to this model, optimal adaptation depends on our ability to confront and transform the dark side of life in service of achieving positive goals. Indeed, the best defense is offence. The most effective way to protect ourselves against the terror of death is to pursue the task of living a meaningful life despite the shadow of death. Both the approach and avoidance systems are needed to be free us from the prison of death fear and to motivate us to be fully engaged with life in whatever little time we have. From this dual-systems perspective, death fear and death acceptance can co-exist and work together for our wellbeing.

Similarly, this model also proposes that sustainable or mature happiness is based on inner peace, equanimity, and harmony, regardless of the circumstances [24], as illustrated by Figure 3. According to Figure 3, mature happiness depends on (1) cultivating inner resources or spiritual virtues, such as courage, wisdom, and meaningful purpose, and (2) the dialectical process of navigating an adaptive balance between self and others and between yin and yang. The resulting mature happiness is also considered noetic happiness because it is based on spiritual virtues.

The solution to avoiding the emotional rollercoaster or getting stuck in the dark pit of painful memories and emotions is to cultivate spiritual resources to restore an abiding sense of calmness and equanimity. Such mature happiness can be best represented by the central point of intersections of all possible human dimensions both horizontally and vertically. Thus, we can have a sense of joy in all kinds of situations, regardless of our race, religion, or social–economic status [25,26].

This spot is the innermost sacred space of calmness, contentment, attunement, harmonious integration, and spiritual blessings. It is also the center point of a cross, symbolizing the paradoxical truth that one needs to go through Hell to reach Heaven, and one needs to lose oneself in order to find the others [27]. Jesus Christ, Buddha, and Lao-tzu are exemplars of such mature happiness.

A number of palliative therapies are consistent with the dual-process model. For instance, the Cancer and Living Meaningfully (CALM) conceptual model in palliative care addresses both the psychological stress and the patient’s need for growth and wellbeing simultaneously [28]. Sessions for CALM address illness/symptoms management while helping the patient maintain a sense of meaning and purpose [29]. Reed’s theory of self-transcendence also addresses the need to turn vulnerability to suffering into personal growth and wellbeing through self-transcendence [30,31].

## 2. Existential Suffering and Palliative Care

### 2.1. A Developmental Perspective

The EPP perspective also enables us to view existential suffering in an entirely new light, as shown in Table 1. Every stage of development, starting with childhood, poses both an existential crisis and opportunity for personal growth. This model complements Erickson’s stage model [32,33] by elaborating the stages of adult development and adding an existential positive psychology dimension. Thus, life is viewed as a constant struggle in every stage of human development. How we resolve each existent crisis may determine whether we enjoy a life of virtues and flourishing or lifelong suffering from the evils and negative consequences of the poor choices we’ve made in completing our developmental tasks.

Towards advancing age and death, we resort to spiritual development and self-transcendence to compensate for losses in physical and mental vitality [6,34]. Wong, Arslan et al. sums up the paradoxical truth of finding meaning in life and death through self-transcendence [27]:
*“Living well and dying well involve enhancing one’s sense of self, one’s relationships with others, and one’s understanding of the transcendent, the spiritual, the supernatural…And only in confronting the inevitability of death does one truly embrace life.”*

### 2.2. Existential Suffering and the Quest for Meaning

With some imagination, I could hear the cry of “Why” from our ancestors: Why has my child been taken by a beast? Why is God angry and why does he punish us with natural calamities? Why is life so hard? Why must we struggle, because at the end we all die? His cry for meaning has never ceased because life is full of suffering and existential crises in every era of history and in every stage of human development.

All people at one time or another have struggled with such existential questions as: Who am I? What am I here for? What should I to do with my life? What really matters in my life? How and where can I find happiness? How can I avoid making the wrong choices or life goals? Where do I belong? Where is my home? What is the point of all my striving? Who am I? 

Older people are more concerned with self-evaluation and regrets [23,35,36]. They may ask questions such as: Have I really lived the life I always wanted? Has my life been worthwhile? What is the point of living in pain? What is the meaning of death? What happens to me after I die? How can I find and give forgiveness? How should I best spend the remaining days of my life? How can I find peace, comfort, and hope in the face of death?

Every cry for meaning serves the dual function of making suffering more bearable and finding a purpose for living; thus, making it more likely for us to move towards a life goal in spite of obstacles and suffering. In real life, most people just go about their daily business until they unexpectedly encounter the horrors of human evil, suffering, and death.

The “Why” questions will push people to search for a reason or purpose for living. One does not need to understand or use the word “meaning” to live a meaningful life. The reality of life and the human nature conspire to nudge us to search for a deeper and more fulfilling way of life.

Peterson [37] argued that human beings are, by nature, religious and spiritual in order to survive in a world full of dangers, evils, and uncertainty. Meaning primarily comes from everyday heroism of taking responsibility for making the necessary sacrifices and aiming at some greater good in the face of widespread evil. Peterson places evil and meaning through the same evolved mechanism of protecting our own vulnerability as finite and fragile human beings in the face of infinite, powerful forces. 

If people have not resolved their existential crises in their early stages of development, their existential crisis becomes more intense at the end, as powerfully portrayed in Tolstoy’s [38]. *The death of Ivan Ilyich*. In the midst of his mental agony and physical pain, Ivan is spiritually reborn in the last hours of his life when he finally realized that his life was not what it should have been. Only after he wants to right the wrong and apologize to his wife, he finds his meaning of life and his existential suffering disappears. Joy fills his heart before he dies.

Another example is classic film *Ikiru* by Akira Kurosawa [39]. In this movie, the protagonist, Watanabe, lives an ordinary life of eating–working–sleeping without reflecting on the meaning of life until cancer strikes. *Ikiru* shows us the existential journey of transforming the terror of dying of cancer into the triumph of meaningful living. Watanabe’s transformation is shown in his final self-sacrificial act of creating a park from a cesspool and his happy death. The process of his good death began with the agonizing search for meaning after his terminal cancer diagnosis.

According to Frankl [40], the medical ministry of logotherapy is an adjunct to the practice of medicine. Patients often struggle with such existential questions, such as: Why me? Why now? Why suffering? Why do I have to die at such a young age? Existential suffering results from one’s inability to find a satisfactory answer to the mysteries of evil, suffering, and death.

In view of the complexity of the search for meaning to resolve one’s existential crisis, Wong [41] proposes a two-factor model of meaning seeking: (1)“Negatively oriented search for meaning—the Why and How questions that increase our ability to understand the cause and reason of unpleasant and unexpected events in order to meet our needs to predict, control and justify them. It represents the lay scientist and lay philosopher in each of us [42,43].(2)Positively oriented search for meaning—the What questions that fulfill our responsibility to do the right thing and pursuing the right life purpose according to our beliefs, conscience, and abilities. It represents the moralist (saint) and idealist (dreamer) in each of us [44,45,46]”.

Palliative care patients are wrestling with both the existential problems of suffering and death, and future meaning to fulfill in their remaining days. For example, Figure 4 lists some of the items in Lo et al.’s [47] Death and Dying Distress Scale.

It is clear from Figure 4 that the search for meaning is mostly negatively oriented meaning seeking, such as regrets for past mistakes and missed opportunities, while some are concerned about unfinished business and seeking future meaning (Items #4 and #5). 

In the Patient Dignity Inventory (PDI) [48], the factor on existential distress is primarily concerned with the absence of meaning in the present, such as “Not feeling worthwhile or valued” and “Feeling life no longer has meaning or purpose”, while the factor on peace of mind is concerned with the need for future meaning, meaningful contribution, and spiritual life.

Therefore, the wellbeing of palliative care patients importantly depends on resolving these existential struggles. According to the Functional Assessment of Chronic Illness Therapy-Spiritual Well-Being [49], the three factors in the spiritual wellbeing subscale essentially depict mature happiness in terms of inner peace and harmony due to spiritual beliefs, faith, reasons for living, and sense of purpose.

### 2.3. The Desire to Hasten Death and Physician-Assisted Suicide

This represents the latest existential issue as physician-assisted suicide gains increasing acceptance. Progress in medicine and technology makes it possible for people to live longer in spite of unbearable suffering and major losses in functionality and all life domains. Theoretically, people could be kept alive almost indefinitely, but at what cost, economically and psychologically? Researchers are beginning to ask whether life is still worth living at 100 years or older [50]. The medical profession is also debating whether physician-assisted suicide should be granted to individuals with mental disorders [51] or physical disabilities [52].

However, research shows that existential beliefs and inner resources [5] can all contribute to wellbeing even at advanced age. In addition, numerous psychological factors can sustain life, even in harsh conditions, such as meaning [40] and valuation of life [53]. It is our belief that good palliative care should include palliative counseling, which deals with psychological and spiritual issues related to suffering and dying, such as religious beliefs [24,54].

Wong and Stiller [55] provided a great deal of information supporting good palliative care as an alternative to euthanasia. They cited Byock [56] as proposing: “we must tell the stories of our patients who have died well, especially those who had previously considered or who might have committed suicide.” (p. 6). Joni Eareckson Tada [57] is such an example. She was a quadriplegic for almost thirty years as a result of a traffic accident. She stated that her ability to minister to others in pain and suffering provided meaning in her life: “The longer I hung in there through the process of suffering, the stronger the weave in the fabric of meaning.” (p. 85). Kübler-Ross [58] also had much to say on physician-assisted suicide:
*“Even with all my suffering I am still opposed to Kevorkian, who takes people’s lives prematurely simply because they are in pain or are uncomfortable. He does not understand that he deprives people of whatever last lessons they have to learn before they can graduate. Right now I am learning patience and submission. As difficult as those lessons are, I know that the Highest of the High has a plan. I know that He has a time that will be right for me to leave my body the way a butterfly leaves its cocoon. Our only purpose in life is growth.”*(p. 281)

## 3. Recent Research on Existential Anxieties and Wellbeing in Palliative Patients

Research in recent years, especially during the COVID-19 pandemic, has illuminated the existential anxieties that palliative care patients face on the daily basis as they approach the end of life. It is clear that these anxieties are multifaceted [59], and the source of much suffering [60]. They still remain a neglected area in palliative care research, especially in everyday palliative care conversations [61].

The level of distress that palliative care patients experience as a result of existential anxieties may be influenced by numerous factors. For example, less distress is associated with older age [62], higher spiritual wellbeing [47,63], attachment security [64,65] and meaning in life [63]. However, previous research has suggested that most, if not all, palliative care patients, as well as their families and care providers, experience some form of existential anxieties [60,66,67]. Some of these anxieties will now be discussed.

### 3.1. Death-Related Anxieties

Death-related anxieties range from practical fear (e.g., process of dying, being a burden to their caregivers and family) to psychosocial or existential fears (e.g., missed opportunities, impact, or burden of death on others). Vehling et al. [68] found that between all the different death and dying related fears within the 15 item Death and Dying Distress Scale (DADDS) [69], fear of “running out of time” was found to be a central concern for palliative care patients. Death-related anxieties are often positively associated with demoralization, characterized by feelings of hopelessness, the loss of meaning, a sense of failure, and meaninglessness [70]; and negatively associated with social relatedness [65].

### 3.2. Grief

There are many forms of grief in palliative care settings. As Moon [71] notes, “Grief and palliative care are interrelated and perhaps mutually inclusive. Conceptually and practically, grief intimately relates to palliative care, as both domains regard the phenomena of loss, suffering, and a desire for abatement of pain burden.” (p. 19). For instance, COVID-19 lockdown procedures, which prevent families from seeing loved ones in palliative care, have led to an increase in anticipatory, disenfranchised, and complicated grief for individuals, families, and medical providers [72]. Families taking care of children that are in Pediatric Palliative Care (PPC) experience anticipatory grief, grief around the time of death, and grief after death [73]. Patients may have lost their spouse before or during palliative care, leading to feelings of numbness, shock, fear, anger, and survivor guilt [74].

### 3.3. Isolation and Loneliness

As a result of their spouse’s death, as well other factors, such as independent living, decline in motor or cognitive abilities, or the recent COVID-19 pandemic, palliative care patients often experience feelings of isolation or loneliness [75,76,77]. Abedini and colleagues [78] found that older individuals who are lonely are more likely to have more illness symptoms (e.g., be troubled by pain, difficulty breathing, or severe fatigue), to use life support in the last 2 years of life, and to die in a nursing home. In their survey across four long term care settings, Sundström and colleagues [79] reported that, in home and residential care, the patients’ existential loneliness was focused on life, both in the past and present, while in hospital and palliative care, existential loneliness was focused on the patient’s coming death. However, high levels of existential loneliness were reported in all four settings. That is why some researchers suggest that now is the time to implement community-hospice settings to help decrease mental health problems that are the result of loneliness [80].

### 3.4. Dignity Related Existential Distress (DR-ED)

Dignity related existential distress (DR-ED) is prevalent among palliative care patients nearing death for a variety of reasons. Bovero and colleagues [81] reported that two factors that accounted for 58% of the DR-ED variance in their study are self-discontinuity and loss of personal autonomy. Self-discontinuity may be a result of declining physical and cognitive functions, preventing patients from taking part in meaningful activities. It may also be a result of taking on new responsibilities as the grandparent of a family or elder in the community. On top of physical and cognitive decline, the patient’s loss of independence and personal autonomy may be from the death of their spouse because elderly couples often maintain their independence by compensating for one another [82].

### 3.5. Regrets

Everyone experiences regret from time to time, whether from mistakes or missed opportunities. However, experiencing regrets at the end of life may intensify end of life suffering because one knows they will be unable to rectify past mistakes [35]. In Bronnie Ware’s [36] bestselling book, she mentioned five common regrets of palliative care patients: wishing that they lived an authentic life, wishing that they did not work so hard, wishing they had the courage to express their feelings, wishing that they stayed in touch with their friends, and wishing they let themselves be happier. Many of these regrets come from an “unlived life”, which may play a role in generating end-of-life death-related anxieties [23].

## 4. Integrative Meaning Therapy (IMT) in Palliative Care

### 4.1. Meaning-Centered Approach to End-of-Life Care

Given the prevalent loss of meaning and dignity in palliative patients, meaning-centered therapies demonstrated efficacy in improving spiritual wellbeing, sense of dignity, and meaning, and decreasing depression, anxiety, and desire for [83,84,85]. A sense of dignity, defined as “quality or state of being worthy, honored, or esteemed” [27], is importantly related to personal meaning. Dignity Therapy [86] is primarily concerned with what has been meaningful in the life of the patients and the legacy patients wants to pass on to family and loved ones.

Another scientifically validated meaning-centered therapy for advanced cancer is Meaning-Centered Psychotherapy [87]. Its aim is to sustain and enhance a sense of meaning in the face of existential crisis. Based on Frankl’s logotherapy, the protocol of Meaning-Centered Psychotherapy [88] is composed of 7–8 sessions in which patients reflect on the concept of meaning and the impact that cancer has produced on their life and identity.

Breitbart’s meaning-centered group therapy for cancer patients [89] aims to help expand possible sources of meaning by teaching the philosophy of meaning, providing group exercises and homework for each individual participant, and by open-ended discussion. The eight group sessions are categorized under the following specific themes of meaning relevant to cancer patients:Session 1—Concepts of meaning and sources of meaningSession 2—Cancer and meaningSessions 3 and 4—Meaning derived from the historical context of lifeSession 5—Meaning derived from attitudinal valuesSession 6—Meaning derived from creative values and responsibilitySession 7—Meaning derived through experiential valuesSession 8—Termination and feedback.

### 4.2. Wong’s Pioneering Work on Death Acceptance

Wong’s integrative meaning therapy is based on his pioneer work on death acceptance.

Elisabeth Kubler-Ross [58] stage-model of coping with death (denial, anger, bargaining, depression, and acceptance) was a milestone in death studies. Approximately 30 years later, Wong and his associates undertook a comprehensive study on death acceptance, which led to the development of the Death Attitudes Profile (DAP) [90] and the Death Attitudes Profile Revised (DAP-R) [91]. Both scales have been widely used worldwide.

In addition to death fear and death avoidance, Wong and associates identified three distinct types of death acceptance: (1) neutral death acceptance—accepting death rationally as part of life; (2) approach acceptance—accepting death as a gateway to a better afterlife; and (3) escape acceptance—choosing death as a better alternative to a painful existence.

Approach acceptance is rooted in religious/spiritual beliefs in a desirable afterlife. To those who embrace such beliefs, the afterlife is more than symbolic immortality, thus offering hope and comfort to the dying as well as the bereaved. Escape acceptance is primarily based on the perception that death offers a welcome relief from the unbearable pain and meaninglessness of staying alive. The construct of neutral acceptance means to accept the reality of death in a rational manner and make best use of one’s limited time on earth.

Approach acceptance may also incorporate neutral acceptance with regard to making the best use of our finite life on earth, but it has the advantage that belief in an afterlife can be a source of comfort and hope in the face of death. Maybe that is why most people believe in heaven or an afterlife [92].

Consistent with Wong’s dual-system model, death anxiety and death acceptance can co-exist. Some form of death anxiety is always present over a wide range of factors, such as ultimate loss, fear of the pain and loneliness of dying, fear of failing to complete one’s life work, the uncertainty of what follows death, annihilation anxiety or fear of non-existence, and worrying about the survivors after one’s death. Even with the constant presence of some level of death anxiety, one can achieve death acceptance through three stages: (1) avoiding death, (2) confronting or facing death, and (3) accepting or embracing death.

### 4.3. Meaning Management and Death Acceptance

Meaning plays an important role in death acceptance because once one has found something worth dying for, one is no longer afraid of death. Meaning-making can help us rise above fear of death and motivate us to strive towards something that is bigger and longer lasting than ourselves. Personally, I (the first author) have gone through the same existential struggles of finding meaning when coping with cancer [93] and loneliness during hospitalization [94].

Wong has described the meaning management theory (MMT) as a conceptual framework to understand and facilitate death acceptance [95]. MMT is based on the existential positive psychology and dual-system model described earlier. According to MMT, it is more productive and fulfilling to confront our death anxiety courageously and honestly and at the same time passionately pursue a meaningful goal [22,96].

MMT is more than cognitive reframing or rationalization. It actually requires a fundamental shift to from pleasure-seeking to the meaning mindset [97], and from self-centeredness to self-transcendence [45]. Meaning therapy [39,98] equips people to squeeze out meaning and hope even from the darkest moments of life.

### 4.4. Some Key Concepts of IMT in Palliative Care

In addition to helping palliative care patients work through their suffering and come to the point of death acceptance through meaning seeking, we want to briefly explain (1) the concept of meaning, (2) the importance of faith, hope, and love, (3) the importance of courage, acceptance, and transformation, and (4) provide a few practical tips for palliative care workers (for details regarding IMT, please read [99]):Wong’s Definition of Meaning. Meaning has been defined by different researchers differently. Wong [39] proposes that a comprehensive way to clarify the concept of meaning is PURE (purpose, understanding, responsibility, and enjoyment):(a)A meaningful life is purposeful. We all have the desire to be significant, we all want our lives to matter. The intrinsic motivation of striving to improve ourselves to achieve a worth goal is a source of meaning (as in the movie *Ikiru*). That is why purpose is the cornerstone of a meaningful life. Even if you want to live an ordinary life, you can still do your best to improve yourself as a good parent, spouse, neighbor, or a decent human being.(b)A meaningful life is understandable or coherent. We need to know who we are, the reasons for our existence, or the reason or objective of our actions [100]. Having a cognitive understanding or a sense of coherence is equally important for meaning.(c)A meaningful life is a responsible one. We must assume full responsibility for our life or for choosing our life goal. Self-determination is based on the responsible use of our freedom. This involves the volition aspect of personality. That is why for both Frankl [44] and Peterson [37] have noted that responsibility equals meaning.(d)A meaningful life is enjoyable and fulfilling. It is the deep life satisfaction that comes from having lived a good life and made some difference in the world. This is a natural by-product of self-evaluation that “my life matters”.

Together, these four criteria constitute the PURE definition of meaning in life. Most meaning researchers support a tripartite definition of meaning in life: comprehension, purpose, and mattering [101,102]. However, these elements are predicated on the assumption that individuals assume the responsibility to choose the narrow path of meaning rather than the broad way of hedonic happiness.

In the existential literature, freedom and responsibility are essential values for an authentic and meaningful life (Frankl, Rollo May, Irvin D. Yalom, Emmy van Deuzen, etc.). For instance, my (the first author) life is meaningful because I chose the life goal of reducing suffering, as well as bringing meaning and hope to suffering people. This was not an easy choice, but it was the only choice if I wanted to be true to my nature and my calling.

2.The Golden Triangle of Faith, Hope, and Love: We have a serious mental health crisis because we are like fish out of water, living in a materialistic consumer society and a digital world without paying much attention to our spiritual needs. Technological progress contributes to our physical wellbeing, but it also destroys the soul if we do not make an intentional effort to care for our soul. IMT aims to help people get back into the water—to meet people’s basic psychological needs for loving relationships, a meaningful life, and faith in God and some transcendental values, as shown in the symbol of the Golden Triangle (Figure 5).

Briefly, faith in God or a higher power represent our spiritual core value: (a)The power of IMT is derived from faith—faith in a better future, in the self, in others, in God, and in a happy afterlife. It does not matter whether you have faith in Jesus, Buddha, or in your medical doctors. If you have faith in someone or something greater than yourself, you will have a better chance of overcoming seemingly insurmountable problems. Faith, nothing but faith, can counteract the horrors of life and death. For Frankl, faith is the key to healing:*“The prisoner who had lost faith in the future—his future—was doomed. With his loss of belief in the future he also lost his spiritual hold; he let himself decline and become subject to mental and physical decay.”*([44], p. 95)(b)Hope represents one’s role as an agent to discover one’s true calling and work towards a better future. Even palliative care patients can work towards a better tomorrow. The saddest thing my (the first author) father said to me during my last visit to Hong Kong was “I have no hope. I’m going to die soon, and none of my children are interested in taking over my business”. This was because he had no hope beyond his own personal interest. Tragic optimism [103,104] enables one to transcend such hopelessness.(c)Love for others and developing connections indicate that we are always part of a larger whole, and relationships are a major source of meaning of life [105]. By withholding love, people perish due to loneliness and meaninglessness. Do we realize that love is the most powerful force on earth? Do we know that love can give us the strength to endure anything, the courage to face any danger, and the joy to make sacrifices for others?

Faith, hope, and love are as essential to our mental health as air, food, and water are essential to our physical health. This positive triad, as depicted by the golden triangle, has enabled humanity to survive since the beginning of time; it is still essential in overcoming depression, addiction, and suffering, and creating a better future. Wellbeing even in palliative care patients, can be conceptualized in terms of the golden triangle.

3.The Iron Triangle of Courage, Acceptance, and Transformation. Life is tough, especially during old age with all the inevitable losses. During the end-of-life stage, one needs a lot of courage to face all the challenges associated with death and dying [106]. One needs courage to cope with the distress of sickness and dying, to accept all the losses, and for the final exit. One also needs courage to connect with their own inner resources, family, and community in order to enhance their dignity and well-being. The main thrust of my (the first author) recent book [107] is that we are wired in such a way that our genes and brain have the necessary capacities to survive and thrive in any adverse situations, provided that we are awakened to our spiritual nature and cultivate our psychological resources. In addition to the golden triangle, our other resources come from the iron triangle of courage, acceptance, and transcendence as shown in Figure 6.

(a)Existential courage is the courage “necessary to make being and becoming possible” ([108], p. 4). As discussed earlier, existential courage is needed in all stages of human development: The courage to embrace the dark side of human existence makes it possible for us make positive changes, to face what cannot be changed or is beyond our control, and to transform all the setbacks and obstacles. The most comprehensive treatment of courage can be found in Yang et al. [109]. They treat courage as a spiritual concept “similar to the existential thoughts of the will to power.” (p. 13). In their words, “To Adler, the will to power is a process of creative energy or psychological force desiring to exert one’s will in overcoming life problems.” (p. 12). Courage is also similar to Frankl’s [44] defiant power of the human spirit.

Courage unleashes the hidden strength and optimism in us to forge ahead in spite of the dangers, obstacles, and sufferings; courage is an attitude of affirmation, of saying “Yes” no matter what, an attitude that has been steeled by prior experience of overcoming adversities and hardships [110].

Existential courage consists of the courage to be true to oneself (authenticity), the courage to belong to a group or serve others (horizontal self-transcendence), and the courage to believe and trust in God or a higher power (vertical self-transcendence). Such existential courage encompasses the three vital connections covered by the golden triangle. In sum, courage is a matter of the heart and the will. It is an attitude of affirmation and optimism that enables one to have the true grit to face whatever life throws at them.

(b)Death acceptance is the other side of life acceptance. David Kuhl [111] writes:

“Do I embrace life, or do I prepare to die? And for all of us, the answers are ultimately similar. Living fully and dying well involve enhancing one’s sense of self, one’s relationships with others, and one’s understanding of the transcendent, the spiritual, and the supernatural. And only in confronting the inevitability of death does one truly embrace life.”

The pathway towards resilience is shown in Figure 7 and explained in Wong [46]. Acceptance is also the first step in facing death anxiety. We are more likely to embrace life when we recognize the spiritual values in the life–death cycle, celebrate the completion of our life’s mission, and live life to the fullest right up to the final moment. There are different levels of acceptance: cognitive, emotional, realistic, existential, and transformative. Death acceptance can transform our life only if we accept it at the deepest existential level. 

(c)There are different pathways in transforming negative events and emotions into wellbeing [112,113]. Transformative coping takes on different forms, such as reframing, re-authoring, or recounting one’s life event in terms of a larger narrative or meta-story. For palliative care patients, life review or reminiscence [42] and small self-transcendental acts, as shown in *Ikiru*, seems most helpful. In life review, we ask patients to reflect on the following questions: What are your happiest moments (with someone special in your life)? What is your best early memory? What are your proudest moments (for your achievement and contribution)? What are your most meaningful moments?

4.Some Practical Tips in Palliative Counselling. Here are some practical tips to help transform a victim’s journey into a hero’s adventure and discover meaning and hope in boundary situations. IMT seeks to awaken the client’s sense of responsibility and meaning, and guide them to (a) achieve a deeper understanding of the problems from a larger perspective and (b) discover their true identity and place in the world.

The transformative approach to spiritual care is based on what you say and do with patients rather than what you do to the patients. “Serving patients may involve spending time with them, holding their hands, and talking about what is important to them.” [8]. It often includes the following elements in a healing relationship: The healing silence—listening to the inner voice.The healing touch—touching the heart and soul.The healing connection—establishing an I–You relationship.The healing presence—providing a caring, compassionate presence.The healing process—nurturing spiritual growth.

Self-transcendence is a natural way to prepare us for finally transcending our physical self and the material world. Lucas [114] is correct that self-transcendence is “the best possible help” for palliative care patients because it broadens their values, opens the door for them to discover something worthy of self-transcendence, and it enables them to find meaning and happiness.

Here are a few methods of engaging in meaningful or self-transcendental activities:(1)Do some random acts of kindness to others.(2)Engage in creative or worthwhile work as a gift to the community or family.(3)Reach out to get reconciled with estranged loved ones.(4)Make a useful contribution to society.(5)Be true to oneself and do something one has always wanted to do.(6)Construct a coherent life story with photos as a legacy to one’s family.

Encourage them to reflect on the following self-affirmations:I believe that life has meaning until my last breath.I am grateful that the reality of suffering and death has showed me what I was meant to be.I can live a happy and meaningful life until my last breath.Life has been very tough but I am grateful that I have overcome its obstacles.I have my regrets but I have found forgiveness.

Practical tips in spiritual care:Show compassion through gentle touch (e.g., holding hand) and smile.Ask them if there is anything you can do for them.Ask them about any concerns (e.g., someone they want to see).Help them see that they have lived a life of meaning and purpose.Assure them their life stories are worth telling and remembering.Assure them they have made a difference in the lives of others.Assure them they can still have hope beyond death through faith.Assure them that they can accept death with inner peace.Offer a prayer if it seems appropriate.

## 5. Conclusions


*“Every society can be best judged by how it treats its vulnerable citizens. The progress of a civilization can be measured not only by technological advancements, but also by progress in the humane treatment of those who cannot help themselves. Therefore, hospice and palliative care represents one of the highest achievements of Christian civilization.”*
[106]

In this paper, we have covered the existential suffering that patients face in palliative care. We have also discussed recent research on existential anxieties and wellbeing in palliative care, as well as existing interventions. Finally, by explaining the IMT approach, we have suggested several strategies to help patients cope with their existential anxiety and to accept death through living a meaningful life.

To provide high quality end-of-life care, professional healthcare caregivers need to come to terms with personal mortality and have resolved their personal existential struggles regarding the meaning and core beliefs of their own lives. When they are aware of their own calling, personal values, beliefs, and attitudes, they create deeper and more significant connections with their patients [54].

Spirituality is at the core of palliative care. Religion and spirituality typically pertain to ultimate meaning and purpose and involves certain spiritual practices or religious beliefs and rituals. Some transcendental beliefs [27] has a sense of sacredness [115]. The quality of palliative care depends on the maturity of the caregiver’s own spirituality. From this point of view, the commitment to the self-care of caregivers and their spiritual development is important. They must become aware of the need to connect first with their source of wellbeing, peace, and personal harmony, with its own spiritual dimension.

As suggested by the healing wheel (Figure 1), good end-of-life care requires teamwork, which may include physicians, nurses, psychologists, pastoral care chaplains, and the community. When the team uses their collective resources and adopts a holistic approach that recognizes the importance of spiritual–existential dimension in patients and their families, it will benefit both healthcare professionals and their patients. We need to learn how to open up our inner life to connect with others at the spiritual level and validate eternal values of shared inner lives. The spiritual connection is equally important as medication. Faith in the shared spiritual dimension of relationships can give hope to the dying. Whenever a patient takes their last breath, our focus shifts instantly from the material world to the eternal hope of the soul.

Since we all have to die anyways, we might as well make this last trip the best trip ever and make our final act the most eloquent statement to who we are and what our lives are all about. To pull this off, we need to start planning now. This may be the best strategy to live and die well.

The main limitation of the present paper is that many of the useful ideas remain to be tested empirically in different cultures. Preferable, the empirical tests will not be limited to individuals, but include the organizational culture and community support because the model presented here is a holistic one.

## Figures and Tables

**Figure 1 medicina-57-00924-f001:**
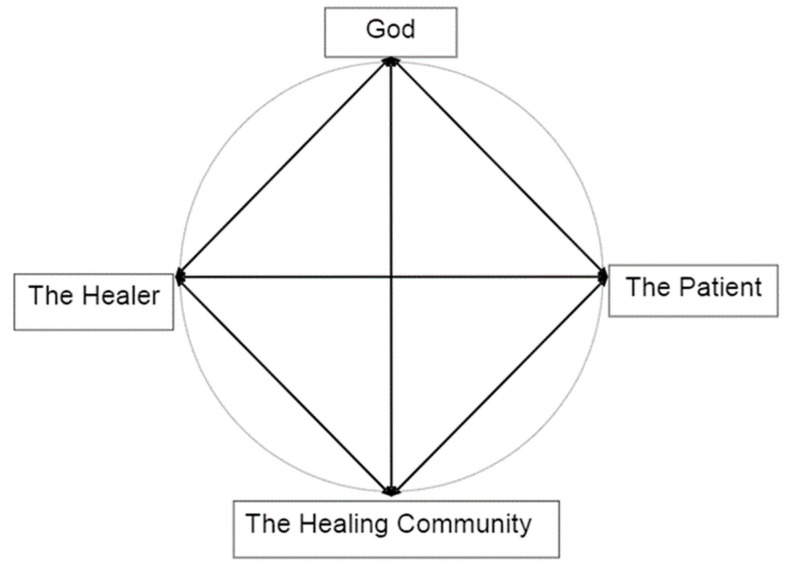
The Healing Wheel in Holistic Health Care.

**Figure 2 medicina-57-00924-f002:**
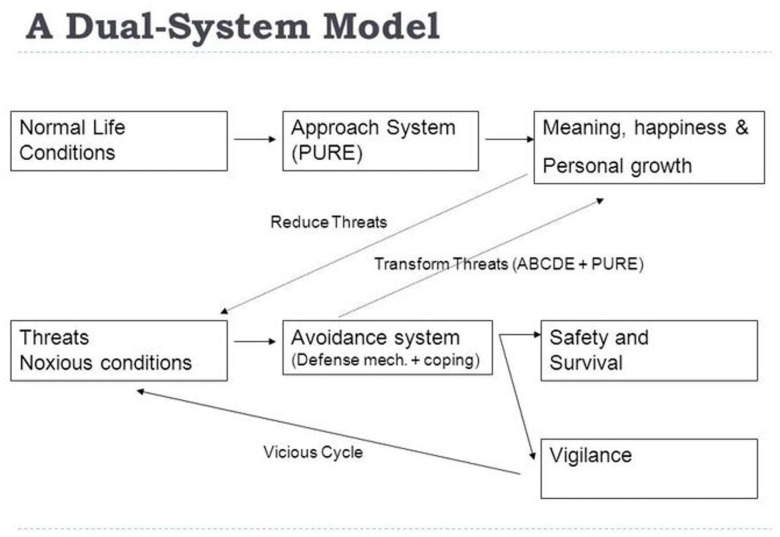
A Dual System Model of What Makes Life Worth Living.

**Figure 3 medicina-57-00924-f003:**
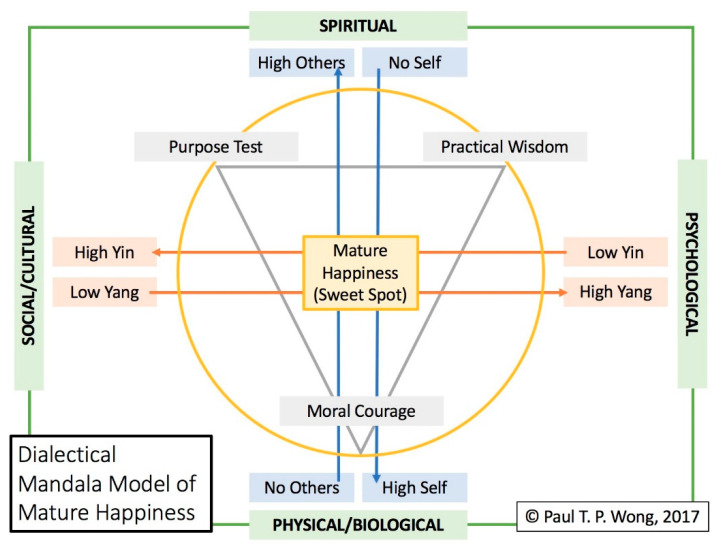
Dialectical Mandala Model of Mature Happiness.

**Figure 4 medicina-57-00924-f004:**
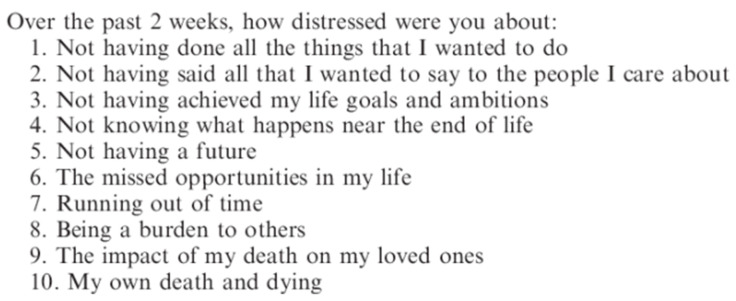
Sample Items from Lo et al.’s Death and Dying Distress Scale.

**Figure 5 medicina-57-00924-f005:**
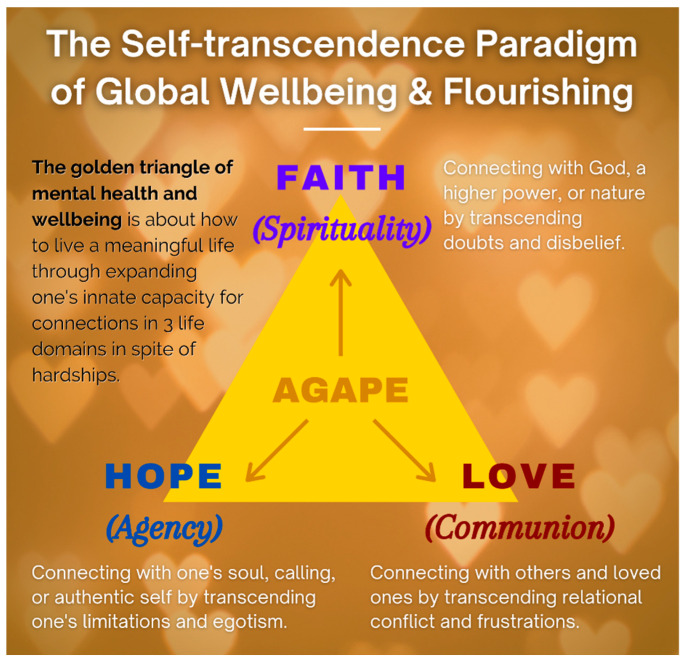
The Golden Triangle.

**Figure 6 medicina-57-00924-f006:**
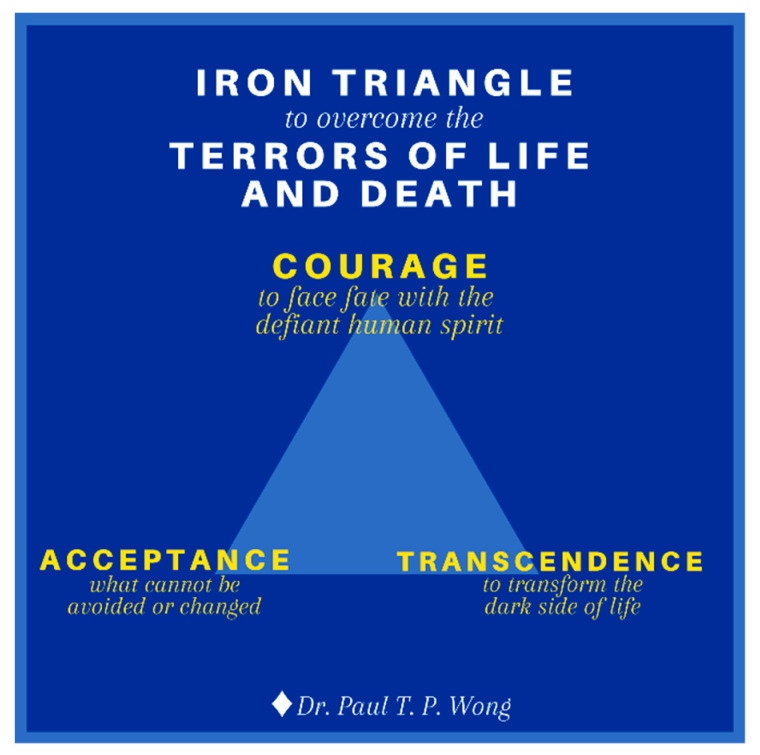
The Iron Triangle.

**Figure 7 medicina-57-00924-f007:**
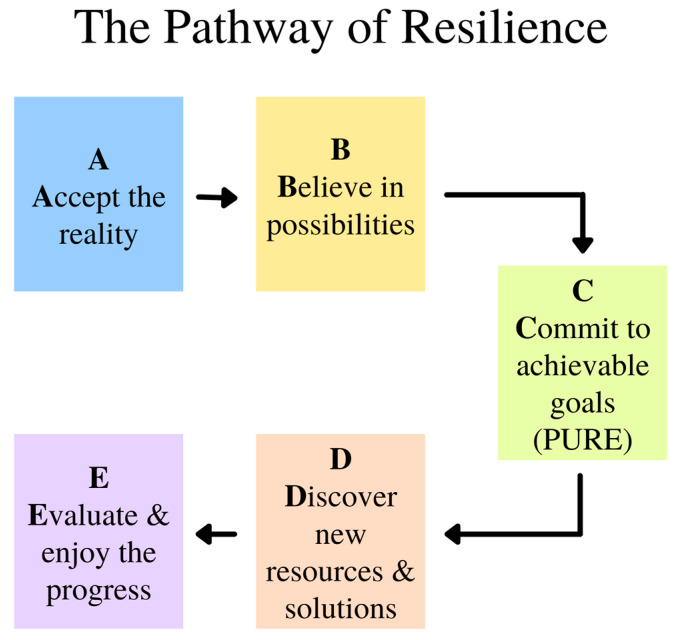
The Pathway of Resilience.

**Table 1 medicina-57-00924-t001:** Different Existential Crisis for Different Stages of Human Development.

Stage	Age	Existential Crisis	Main Task	Gains	Risks
Infancy	Birth–2 years	Separation anxiety	Necessary gradual separation from mother	- Trust, faith, and hope - Secure attachment (love) - Enduring discomfort - Delayed gratification	- Dependency - Anxious/avoidant attachment - No frustration tolerance - Narcissism - Fear of abandonment
Preschooler	3–4 years	Safety anxiety (Fear of getting hurt)	Testing limits of autonomy	- Obedience - Freedom and security - within boundaries - Honesty and speaking the truth - Respect for rules and authority	- No respect for parents and authority - No impulse control - Exerting power through temper tantrums - Deception - Aggression
Kindergarten to primary school	4–12 years	Social anxiety (Fear of not belonging)	School	- Sharing and belonging - Playing fair (justice) - Humility and forgiveness - Curiosity about the world	- Isolation/loneliness - Social anxiety - Fear of rejection - Bullying and cruelty - Poor self-esteem - Manipulation
Adolescence	12–18 years	Identity crisis	- Puberty - Preparation for adulthood	- Self-knowledge - Self-awareness - Sexual orientation - Discovery of areas of strengths	- Role confusion - Dropping out of school - Low achievement motivation - Seeking pleasure and risky behaviour - Rebelliousness and antisocial behaviour
Young adult or early career	19–25	Independence anxiety	- Love relationship - Entry into work force	- Courage - Hope - Purpose - Confidence in love relationship - and work	- No meaning and purpose - Depression - Aggressiveness - Addiction - Loner - Making a living through illegal means
Adult or mid-career	25–40	Achievement anxiety (Fear of failure in career and marriage)	- Supporting a family - Parenting	- Responsibility - Resourcefulness - Perseverance - Career success - Happy marriage - A sense of actualization	- Getting stuck in a bad job or bad marriage - Divorce - Delinquent children - No close friends - Depression and addiction
Mature adult or late career	40–60	Mid-life crisis	- Reflection on the first half of life - Ready for major change	- Generativity - Life satisfaction - Life transformation - Social conscience - Consolidating one’s - contributions - Redemption	- Stagnation - Regression to adolescence - Taking unwise risks - Taking early retirement - Giving up on life
Early old age	60–75	Ultimate concerns about boredom and meaninglessness	Retirement	- Self-transcendence - Integrity - Spiritual growth - Enjoying life to the fullest - Volunteering - Grand-parenting	- Despair - Depression - Bitterness - Resentment - Blaming and complaining - Cranky old person
Late old age	76–death	Worrying about unfinished business	Completing the race gracefully	- Letting go, facing death with - gratitude and faith - Integration - Death acceptance - Legacy - Hope for immortality - Wisdom - Spiritual maturity - Mature happiness	- Regrets - Despair - Depression - Anger towards life - Suicide - Alienating adult children

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
