# Peer review of "Existential Suffering in Palliative Care: An Existential Positive Psychology Perspective"

_medicina, 2021, doi:10.3390/medicina57090924_

Round 1
Reviewer 1 Report
Hello Authors,
I would like to see more structure to your paragraphs and improvement on the wording. For example, using words like inadequacies and laid bare in the same sentence seems to be confusing. I understand what you are trying to portray though. In addition, if you strengthen the theoretical framework with more updates references it will improve the manuscript. Thank you.
Author Response
Q: I would like to see more structure to your paragraphs and improvement on the wording. For example, using words like inadequacies and laid bare in the same sentence seems to be confusing. I understand what you are trying to portray though.
A: Thank you for your suggestion. We have improved the structure and wording of the paragraphs. Please see the tracked changes in the revised paper.
In addition, if you strengthen the theoretical framework with more updates references it will improve the manuscript.
A: Thank you for your suggestion. We have strengthened the theoretical framework with more references, especially on page 6.
Reviewer 2 Report
Thank you for the opportunity to review this manuscript.
Recommendations
Please cited the references according with the rules of the journals.
Figure 3 has not enought explanation related to the topic.
I recommend adding the limitations and the strenght point of your study.
The conclussions need to be revised, its focus on the main ideas related to the topic.
Author Response
Q: Please cited the references according with the rules of the journals.
A: Thank you for your suggestion. The authors of this paper come from a psychology background, and are therefore more comfortable with APA than ACS. We have contacted the Medicina editorial office about this. They are okay with APA references.
Q: Figure 3 has not enought explanation related to the topic.
A: Thank you for your suggestion. We have replaced Figure 3 with a more detailed figure related to mature happiness. See page 9.
Q: I recommend adding the limitations and the strenght point of your study.
A: Thank you for your suggestion. We have highlighted the limitations in the last paragraph on page 33: “The main limitation of the present paper is that many of the useful ideas remain to be tested empirically in different cultures. Preferable, the empirical tests will not be limited to individuals, but include the organizational culture and community support because the model presented here is a holistic one.”
Q: The conclussions need to be revised, its focus on the main ideas related to the topic.
A: Thank you for your suggestion. We have included a paragraph briefly summarizing the main ideas of the paper.